# Synthesis, Characterization, and Antifungal Activity of Schiff Bases of Inulin Bearing Pyridine ring

**DOI:** 10.3390/polym11020371

**Published:** 2019-02-20

**Authors:** Lijie Wei, Wenqiang Tan, Jingjing Zhang, Yingqi Mi, Fang Dong, Qing Li, Zhanyong Guo

**Affiliations:** 1Key Laboratory of Coastal Biology and Bioresource Utilization, Yantai Institute of Coastal Zone Research, Chinese Academy of Sciences, Yantai 264003, China; ljwei@yic.ac.cn (L.W.); wqtan@yic.ac.cn (W.T.); jingjingzhang@yic.ac.cn (J.Z.); yqmi@yic.ac.cn (Y.M.); fdong@yic.ac.cn (F.D.); 2University of Chinese Academy of Sciences, Beijing 100049, China

**Keywords:** inulin, Schiff base, antifungal

## Abstract

As a renewable, biocompatible, and biodegradable polysaccharide, inulin has a good solubility in water and some physiological functions. Chemical modification is one of the important methods to improve the bioactivity of inulin. In this paper, based on 6-amino-6-deoxy-3,4-acetyl inulin (**3**), three kinds of Schiff bases of inulin bearing pyridine rings were successfully designed and synthesized. Detailed structural characterization was carried out using FTIR, ^13^C NMR, and ^1^H NMR spectroscopy, and elemental analysis. Moreover, the antifungal activity of Schiff bases of inulin against three plant pathogenic fungi, including *Botrytis cinerea*, *Fusarium oxysporum* f.sp.*niveum*, and *Phomopsis asparagi*, were evaluated using in vitro hypha measurements. Inulin, as a natural polysaccharide, did not possess any antifungal activity at the tested concentration against the targeted fungi. Compared with inulin and the intermediate product 6-amino-6-deoxy-3,4-acetyl inulin (**3**), all the synthesized Schiff bases of inulin derivatives with >54.0% inhibitory index at 2.0 mg/mL exhibited enhanced antifungal activity. 3NS, with an inhibitory index of 77.0% exhibited good antifungal activity against *Botrytis cinerea* at 2.0 mg/mL. The synthesized Schiff bases of inulin bearing pyridine rings can be prepared for novel antifungal agents to expand the application of inulin.

## 1. Introduction

In developing countries, there are a large number of economic losses caused by plant pathogenic fungi every year [1]. It has been reported that *Botrytis cinerea (B. cinerea)* can cause the grey mold disease which could cause fruits and vegetables to rot after harvesting [2]. *Fusarium oxysporum* f.sp.*niveum* (*F. oxysporum* f.sp.*niveum*) can cause Fusarium wilt of watermelon [3,4]. *Phomopsis asparagi (P. asparagi)* can blight the asparagus stems, which has been considered as a great threat to asparagus [5]. In general, chemical pesticides were widely used to control pathogenic diseases. However, abuse of chemical fungicides causes harm to the environment and human health [6]. The demand for developing efficacious, environmentally friendly, and natural alternatives has been growing in recent years. 

Inulin is usually obtained from low-requirement crops, such as Helianthus tuberosus, chicory, yacon, and so on [7]. It primarily consists of β (2→1)-fructosyl fructose units (Fm) and a glucopyranose unit at the reducing end (GFn) [8]. As a kind of natural and biodegradable polysaccharide, inulin is a prebiotic ingredient and is being increasingly used in food as a fat alternative [9]. As a renewable source, inulin also has some advantages in unique physicochemical characteristics and biological effects, such as non-toxicity, biodegradability, biocompatibility, being liquid phase adsorption, anticancer properties, immunomodulatory, and so on, which suggests that inulin can be widely applied in the pharmaceutical industry [10,11,12,13,14,15]. However, the low bioactivity of inulin limits its application. Many efforts have been devoted to the chemical modification of inulin in order to improve its bioactivity. Stevens reported a relatively integral overview of the chemical modifications of inulin and their industrial applications [16]. According to the concerned literature, most of the chemical modifications were through ester or ether bonds, since there are only hydroxyl groups in the molecule of inulin [17]. Among all the chemically modified inulin derivatives, the successful synthesis of 6-amino-6-deoxyinulin with the primary amino groups provided more possibilities for the chemical modifications of inulin [18].

Pyridine, as an important aromatic heterocyclic organic solvent and reagent, is used as a pharmacophore for agrochemicals [19,20]. Pyridine and its derivatives have been introduced into all kinds of polysaccharides in order to improve their solubility, physicochemical, and biological properties, which can be applied in mental absorption, gene carrier, antimicrobial, sensor, and biomedical areas [19,21,22,23,24,25]. Jia reported that the solubility and the antifungal activity of the pyridine-grafted chitosan derivatives were improved, which could be used as antifungal agents in the food industry with non-toxicity [19]. Besides, pyridine was introduced into starch to improve its antifungal activity [26]. Hu reported a series of inulin derivatives with amino-pyridines based on chloracetyl inulin, which showed better antioxidant activities than inulin [27]. 

Schiff bases, containing the C=N group, are formed by the reaction between the primary amino and the active carbonyl groups [28]. Schiff bases have been widely applied in catalysts, pharmacology, and antifungal areas [29,30,31,32]. It has been found that the antifungal activities of Schiff bases of chitosan were obviously better than that of acylated chitosan [33]. The Schiff bases with an aromatic nucleus usually have good bioactivities. Guo reported that Schiff bases of 4-amino-pyridine were introduced into inulin by reacting with chloroacetyl and the products showed better antifungal activity than inulin [7]. However, the toxicities and residues of chlorine atoms would threaten the environment and human health. In this paper, a series of Schiff bases of inulin-bearing pyridine based on 6-amino-6-deoxyinulin were successfully synthesized to obtain efficient and environmentally friendly antimicrobial agents. The chemical structures of the inulin derivatives were determined using FTIR, ^13^C NMR, and ^1^H NMR. In addition, three kinds of plant pathogens, including *B. cinerea* (ATCC48340), *F. oxysporum* f.sp.*niveum* (ATCC 66054), and *P. asparagi* (ATCC24625), were selected to evaluate the antifungal activity of inulin and the inulin derivatives by hypha measurements in vitro. 

## 2. Materials and Methods

### 2.1. Materials

Inulin (MW 10000) was purchased from Xi’an Baichuan Biotechnology Co., Ltd (Xi’an, China). *N*,*N*-dimethylformamide (DMF), *N*-bromosuccinimide (NBS), triphenylphosphine (Ph_3_P), sodium azide (NaN_3_), dimethyl sulfoxide (DMSO), pyridine-2-carboxaldehyde, pyridine-3-carboxaldehyde, pyridine-4-carboxaldehyde, and acetic anhydride were supplied by Sinopharm Chemical Reagent Co., Ltd. (Shanghai, China).

### 2.2. Analytical Methods

Fourier Transform Infrared (FTIR) spectra were recorded on a Jasco-4100 ranging from 4000 to 400 cm^−1^ (Japan, provided by JASCO Co., Ltd., Shanghai, China) at 25 °C. The samples were mixed with potassium bromide (KBr) by 1:100 and scanned using the transmittance mode with accumulation of 16 scans and resolution of 4.0 cm^−1^. Nuclear magnetic resonance (^13^C NMR and ^1^H NMR) spectra were recorded on a Bruker AVIII 500 spectrometer (Fällanden, Switzerlnd, provided by Bruker Biospin CN/Bruker (Beijing) Tech. and Serv. Co., Ltd., Beijing, China), using DMSO-*d*_6_ as solvents with tetramethylsilane (TMS) as the internal standard. Chemical shift values were given in δ (ppm). Elemental analysis (C, H, and N) were performed on a Vario Micro Elemental Analyzer (Elementar Trading (shanghai) Co. Ltd., Shanghai, China). The degree of substitution (DS) of inulin derivatives was evaluated on the basis of ^13^C NMR and elemental analysis. The formula for calculating the degree of substitution is as follows:
$$\mathrm{DS}2=\frac{\left(n1+n2\times \mathrm{DS}1\right)\times \mathrm{Mc}}{W1\times \mathrm{Mn}\times n3}$$
$$\mathrm{DS}3=\frac{\mathrm{Mc}\times \left(n1+n2\times \mathrm{DS}1\right)-\mathrm{Mn}\times n3\times W2\times \mathrm{DS}2}{W2\times \mathrm{Mn}\times \left(n4-n3\right)}$$
$$\mathrm{DS}4=\frac{\left(n1+n2\times \mathrm{DS}1\right)\times \mathrm{Mc}-W3\times \mathrm{Mn}\times \left[n3\times \mathrm{DS}2+\left(n4-n3\right)\times \mathrm{DS}3\right]}{W3\times \mathrm{Mn}-n5\times \mathrm{Mc}}$$where DS1, DS2, DS3, and DS4 represent the degree of substitution of 6-bromo-6-deoxy-3,4-acetyl inulin (**1**), 6-azido-6-deoxy-3,4-acetyl inulin (**2**), 6-amino-6-deoxy-3,4-acetyl inulin (**3**), and Schiff bases of inulin, respectively. DS1 is calculated based on the integration areas of carbon atoms in the ^13^C NMR spectra, DS1 = 1.17. Mc and Mn are the molar mass of carbon and nitrogen, Mc = 12, Mn = 14; n1, n2, and n5 are the number of carbon of inulin, diacetyl group, and Schiff bases bearing pyridine ring, n1 = 6, n2 = 4; n5 = 6; n3 and n4 are the number of nitrogen of azido group and amino group, n3 = 3, n4 = 1; W1, W2, and W3 represent the mass ratio between carbon and nitrogen in the inulin derivatives.

The results were processed by computer programs Excel (Microsoft, Redmond, WA, USA), OriginPro 8 (OriginLab, Northampton, MA, USA), R Project (MathSoft, Bothell, WA, USA), and MestReNova (Mestrelab Research S.L., Santiago de Compostela, Spain).

### 2.3. The Synthesis of the Inulin Derivatives

#### 2.3.1. Synthesis of 6-bromo-6-deoxy-3,4-acetyl Inulin (**1**)

The synthetic strategy for the inulin derivatives is shown in Scheme 1. The synthesis of 6-bromo-6-deoxy-3,4-acetyl inulin (**1**) has been published previously [34]. Inulin (4.8 g, 30.0 mmol) were dissolved in anhydrous DMF 50.0 mL, then NBS (16.0 g, 90.0 mmol) and Ph_3_P (23.6 g, 90.0 mmol) were slowly added in an ice bath. The mixture was stirred at 80 °C under the argon protection. Then the mixture was precipitated in extra acetone. The precipitate was collected by filtration and washed three times with acetone. Then, the above precipitates were dissolved in 50.0 mL anhydrous pyridine solution, and acetic anhydride (16.8 mL, 180.0 mmol) was added to the mixture. The solution was stirred at room temperature for 12 h under the argon protection. The precipitate was isolated by the addition of ice water. The product was collected by filtration and washed with ice water. After being dialyzed against deionized water for 48 h to remove the impurities, 6-bromo-6-deoxy-3,4-acetyl inulin (**1**) was obtained by freeze-drying. Yield: 86.8%, DS: 1.17 (^13^C NMR).

#### 2.3.2. Synthesis of 6-azido-6-deoxy-3,4-acetyl Inulin (**2**)

The synthesis of 6-azido-6-deoxy-3,4-acetyl inulin (**2**) was adapted from previous work [18]. In a 250.0 mL round-bottom flask, 6-bromo-6-deoxy-3,4-acetyl inulin (**1**) (2.6 g, 10.0 mmol) was dissolved in 100.0 mL DMF, then NaN_3_ (1.95 g, 30.0 mmol) was added into the solution. After being stirred at 80 °C for 4 h under the argon protection, the mixture was poured into extra distilled water, and the product was purified by dialysis for 48 h. Then, 6-azido-6-deoxy-3,4-acetyl inulin (**2**) was obtained by freeze-drying. Yield: 85.3%, DS: 1.08 (Table 1).

#### 2.3.3. Synthesis of 6-amino-6-deoxy-3,4-acetyl Inulin (**3**)

A mixture of 6-azido-6-deoxy-3,4-acetyl inulin (**2**) (2.7 g, 10.0 mmol) and Ph_3_P (7.9 g, 30.0 mmol) was dissolved in 100.0 mL DMF with stirring at 60 °C for 12 h. The resulting mixture was cooled to room temperature and then was poured into the mixture solution of diethyl ether and ethanol. The precipitate was collected by filtration and washed with ethanol. After being dialyzed against deionized water for 48 h to remove the probably remained impurities, 6-amino-6-deoxy-3,4-acetyl inulin (**3**) was obtained by freeze-drying. Yield: 83.9%, DS: 1.10 (Table 1).

#### 2.3.4. Synthesis of Schiff Bases of Inulin Bearing Pyridine Rings (2NS, 3NS, and 4NS)

6-amino-6-deoxy-3,4-acetyl inulin (**3**) (2.5 g, 10.0 mmol) was dissolved in 30.0 mL DMSO, and then 30.0 mmol various pyridylaldehydes were added into the solution. The mixture was magnetically stirred at 60 °C for 12 h. The solution was stratified in diethyl ether. The lower layer was poured into extra acetone. The precipitate was filtrated and washed three times with enough acetone. After being dialyzed against deionized water for 48 h, Schiff bases of inulin bearing pyridine rings (2NS, 3NS, and 4NS) were obtained by freeze-drying. 2NS, Yield: 83.5%, DS: 0.32; 3NS, Yield: 87.0%, DS: 0.39; 4NS, Yield: 84.8%, DS: 0.56 (Table 1).

### 2.4. Antifungal Assay

The antifungal assay was evaluated against *B. cinerea*, *F. oxysporum* f.sp.*niveum*, and *P. asparagi* in vitro by measuring the growth rate of mycelium according to the method of Guo [35]. Briefly, the compounds (inulin and the inulin derivatives) were dissolved in distilled water at a concentration of 6.0 mg/mL at room temperature. Then, the test sample solution was added to the sterilized potato dextrose agar (PDA) medium to get a final concentration of 0.1, 0.5, 1.0, and 2.0 mg/mL, respectively, and then the solution was poured into the sterilized Petri dishes (6.0 cm). Identical volume distilled water substituting samples were poured into control plates. Finally, the fungi mycelia disk with a diameter of 5.0 mm was placed into the center of the PDA Petri dishes and incubated at 27 °C for 2–3 days. When the diameter of the fungi mycelium reached to the edges of the control plate (without the sample), the inhibitory index was calculated as follows:
Inhibitory index (%)=(1−Da÷Db)×100where Da is the diameter of the growth zone in the test plates, and Db is the diameter of the growth zone in the control plate. The experiments were performed three times, and all the data were averaged and expressed as means ± SD (*n* = 3). 

## 3. Results and Discussion

### 3.1. Structure of Schiff Bases of Inulin

The structures of inulin and the inulin derivatives were determined using FTIR, ^13^C NMR, and ^1^H NMR spectroscopy. The FTIR of inulin and all the inulin derivatives was shown in Figure 1. As shown in Figure 1, the spectrum of inulin showed that the saccharide mainly contains the following characteristic band: 3385 cm^−1^, 1030 cm^−1^, and 868 cm^−1^ [36]. In the 6-bromo-6-deoxy-3,4-acetyl inulin (**1**), the new peaks at 1735 cm^−1^ and 653 cm^−1^ were assigned to C=O and C–Br, respectively. After the chemical reaction between 6-bromo-6-deoxy-3,4-acetyl inulin and NaN_3_, a new strong peak at 2107 cm^−1^ was attributed to the stretching vibration of the azido group [18]. After reduction of the azido group using Ph_3_P, the peak at 2107 cm^−1^ disappeared in 6-amino-6-deoxy-3,4-acetyl inulin (**3**). Meantime, the new peak at 1593 cm^−1^ was assigned to the absorption of NH_2_ [18]. In the spectra of Schiff bases of inulin, except for the carbonyl peak at 1740 cm^−1^, the new peaks at 1610, 1538, 828 cm^−1^, 1612, 1538, 822 cm^−1^, and 1600, 1547, 820 cm^−1^ were assigned to the pyridine rings in 2NS, 3NS, and 4NS, respectively [7]. The characteristic peak of C=N was observed at 1667 or 1666 cm^−1^ [7]. The above results preliminarily demonstrated that Schiff bases of inulin were obtained.

Figure 2 shows the ^13^C NMR spectra of inulin and 6-bromo-6-deoxy-3,4-acetyl inulin (**1**). The signals above 60.0 ppm were assigned to the chemical shift of ^13^C NMR of inulin [18,37]. After reacting with acetic anhydride, there were new peaks at 170 ppm and 22 ppm in **1**, which were related to the carbon of C=O and COCH_3_. The degree of substitution of 6-bromo-6-deoxy-3,4-acetyl inulin (**1**) was evaluated on the basis of the integral values of the ^13^C NMR spectrum in Figure 2 [36]. The formula to determine DS1 of the inulin derivative **1** is shown in the following equation:
$$\mathrm{DS}1=\frac{A}{B}$$where A represents the integration areas of carbons in CH_3_ group of inulin derivative **1**, and B represents the integration areas of carbons of at C_2_ (δ = 98.7 to 101.1 ppm) of furanose rings. 

The ^1^H NMR spectra of inulin and all Schiff bases of inulin are shown in Figure 3. It has been known that the peaks at δ = 3.0–5.2 ppm were assigned to the absorption peaks of protons in the fructose and glucose skeleton of inulin [38]. The peak at δ = 5.2 ppm was assigned to the α-anomeric forms of free glucose [37]. The signal of protons at 1.8 ppm was assigned to the primary amino groups in **3** [18]. In 2NS, 3NS, and 4NS, multiple peaks at δ = 7.7–8.6 ppm were assigned to the pyridine ring and CH=N [7]. Besides, the peak at δ = 2.1 ppm revealed the presence of the CH_3_–C=O group. The absorption peak at δ = 1.9 ppm indicated the residual amino group. The ^1^H NMR spectra further confirmed the successful synthesis of Schiff bases of inulin.

### 3.2. Antifungal Activity

Inulin, 2NS, 3NS, and 4NS all had good solubility in water (Figure 4) and were prepared as solutions at room temperature. The antifungal activities of inulin, 6-amino-6-deoxy-3,4-acetyl inulin (**3**), 2NS, 3NS, and 4NS against *B. cinerea*, *F. oxysporum* f.sp.*niveum*, and *P. asparagi* were investigated by measuring the growth rate of mycelium in vitro. The concentration of all the samples was prepared from 0.1 to 2.0 mg/mL. Carbendazim was used as a positive control in this study. The antifungal indices are shown in Figure 5, Figure 6 and Figure 7. 

Figure 5 shows the inhibitory indices of inulin, 6-amino-6-deoxy-3,4-acetyl inulin (**3**), 2NS, 3NS, and 4NS against *B. cinerea* at all the tested concentrations. According to the graph, we concluded the results as follows. As a positive control, carbendazim with an inhibitory index of 100% could totally inhibit the growth of *B. cinerea*, even at 0.1 mg/mL. However, inulin didn’t show any antifungal activity even at 2.0 mg/mL, which was in accordance with the earlier report [7]. Besides, the antifungal indices of the inulin derivatives was enhanced upon increasing the concentration. The inhibitory index of 6-amino-6-deoxy-3,4-acetyl inulin (**3**) was 69.5% at 2.0 mg/mL. It has been reported that the amino group could contribute to the antifungal activity since the amino group would interact with anionic components of the cell membranes, such as glucan, mannan, proteins, and lipids, to destroy the cell membranes or to form an impervious layer preventing the transport of essential nutrients from entering the cell [18]. Besides, the inhibitory indices of Schiff bases inulin 2NS, 3NS, and 4NS were 72.2%, 77.0%, and 69.9% at 2.0 mg/mL, respectively. It was obvious that all Schiff bases of inulin bearing pyridine rings exhibited excellent antifungal activity compared with inulin, especially at 2.0 mg/mL. The aromatic moieties as favorable factors could contribute to the antifungal activity [39]. Compared with 6-amino-6-deoxy-3,4-acetyl inulin (**3**), the Schiff bases of inulin, such as 3NS, showed relatively better antifungal activity. In addition, the antifungal activity of Schiff bases of inulin with the different position of the nitrogen atom on the pyridine ring didn’t show significant difference.

The inhibitory indices of inulin and the inulin derivatives against *P. asparagi* and *F. oxysporum* f.sp.*niveum* are shown in Figure 6 and Figure 7, respectively. Carbendazim also exhibited prominent antifungal activity against *P. asparagi* and *F. oxysporum* f.sp.*niveum* even at 0.1 mg/mL. Figure 6 shows the antifungal activity of inulin, 6-amino-6-deoxy-3,4-acetyl inulin (**3**), 2NS, 3NS, and 4NS against *P. asparagi*. The results were almost similar to the antifungal activity against *B. cinerea*. Firstly, inulin as a nutrition polysaccharide had no marked inhibition of fungi growth even at 2.0 mg/mL. Secondly, the inhibitory indices of all the inulin derivatives were in a concentration-dependent manner. Thirdly, the compound 3 and all Schiff bases of inulin exhibited enhanced antifungal activity than inulin especially at 2.0 mg/mL, which was attributed to the introduction of the active group such as the amino group and the pyridine ring. Besides, the Schiff bases of inulin especially 2NS and 3NS with inhibitory indices >67% showed relatively better antifungal activity than 6-amino-6-deoxy-3,4-acetyl inulin (**3**). The antifungal activities of inulin and the inulin derivatives against *F. oxysporum* f.sp.*niveum* are shown in Figure 7. The results were similar to the above results against *B. cinerea* and *P. asparagi*. Inulin didn’t show any antifungal activity against *F. oxysporum* f.sp.*niveum* under the tested condition. Besides, the inhibitory indices of the inulin derivatives mounted up with the increasing concentration. The inhibitory indices of 6-amino-6-deoxy-3,4-acetyl inulin (**3**), 2NS, 3NS, and 4NS could reach 52.4%, 57.4%, 70.9%, and 66.5% at 2.0 mg/mL, respectively. All the inulin derivatives showed more enhanced antifungal activity than inulin, especially at 2.0 mg/mL. Besides, as illustrated by the data, the introduction of Schiff bases was more significant for enhancing the antifungal activity of inulin.

Based on the results mentioned above, all Schiff bases of inulin enhanced the antifungal activity compared with inulin and 6-amino-6-deoxy-3,4-acetyl inulin (**3**), especially at 2.0 mg/mL. The introduction of the pyridine ring was an important factor to improve the antifungal activity. The lipophilic characteristic of the heterocyclic aromatic ring will be more likely to pass through the oil film outside the cell membranes based on the “like dissolves like” point, which will damage the cell membranes. Besides, there was no significant difference in antifungal activity between 2NS, 3NS, and 4NS. The inhibitory index showed extremely significant differences among the samples and the concentrations (*p* = 2e^−16^ < 0.01, *p* = 2e^−16^ < 0.01, two-way Analysis of Variance (ANOVA) test). Further analysis showed that the differences between samples were significant (*p* < 0.05, Tukey test). The differences between concentrations were extremely significant (*p* < 0.01, Tukey test).

## 4. Conclusions

In this study, a series of new Schiff bases of inulin were successfully designed, synthesized, and characterized using FTIR, ^13^C NMR, and ^1^H NMR spectroscopy, and elemental analysis. The antifungal activity against three kinds of phytopathogen, including *B. cinerea*, *F. oxysporum* f.sp.*niveum*, and *P. asparagi* were estimated using in vitro hyphal measurements. All Schiff bases of inulin had good solubility in water and showed enhanced antifungal activity when compared with inulin, especially at 2.0 mg/mL. The results indicated that the pyridine group should be the antifungal function group. The different position of *N* atom on the pyridine ring didn’t show significant influence on the antifungal activity. The relationship between the structure and the activity needs further study in the future.

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
