# Peer review of "Synthesis, Characterization, and Antifungal Activity of Schiff Bases of Inulin Bearing Pyridine ring"

_polymers, 2019, doi:10.3390/polym11020371_

Round 1

Reviewer 1 Report

The manuscript of Zh. Guo et al “Synthesis, characterization, and antifungal activity of Schiff bases of inulin bearing pyridine ring” is devoted to synthesis, characterization and antifungal activity of novel 6-amino-6-deoxy-3,4-acetyl-inulin derivatives bearing pyridine ring (2NS, 3NS, and 4NS). The work is very interesting and useful. The characterization of novel inulin’s derivatives was provided by different methods including FTIR, 13C NMR, 1H NMR spectroscopy and elemental analysis. The antifungal activity of novel Schiff bases of inulin against three plant pathogenic fungi was evaluated using in vitro methods.

Unfortunately, the manuscript could be not accepted for publication in this form due to following mistakes:

1.        For determination of degree of substitution (DS) authors of (1) are used the correlation of integration areas of carbon atoms of methyl groups and C2 atom of furanose ring (lines 158-166). To my opinion it’s a wrong idea due to big differences in relaxation time of carbon atoms with different chemical environments. Much more better results of DS may be obtained using 1H NMR spectroscopy.

2.       It is unclear how was calculated the DS for other compounds. The DS calculated using C/N ratio (table 1) is strongly differ for those presented in table 1. For example, the DS for compounds 2NS reveals 0.95 vs 0.30 in table 1.

3.       As a positive control was used carbendazim – compounds of different class benzimidazole. It should be explained.

4.       The inhibitory activities of 2NS, 3NS, 4NS are compared with inulin. However, to my opinions the inhibitory activities of compound 3 should be included because it is well known that amines and ammonium salts are effective antifungal agents.

5.       Lines 165,166. “Protons” should be replaced for “carbons”.

6.       Line 173. The peak at 1.9 ppm could be not assigned to residual amino group. It should be proved. The 1H NMR spectra of compound 3 should be provided.

Author Response

Reviewer 1

the manuscript of Zh. Guo et al “Synthesis, characterization, and antifungal activity of Schiff bases of inulin bearing pyridine ring” is devoted to synthesis, characterization and antifungal activity of novel 6-amino-6-deoxy-3,4-acetyl-inulin derivatives bearing pyridine ring (2NS, 3NS, and 4NS). The work is very interesting and useful. The characterization of novel inulin’s derivatives was provided by different methods including FTIR, 13C NMR, 1H NMR spectroscopy and elemental analysis. The antifungal activity of novel Schiff bases of inulin against three plant pathogenic fungi was evaluated using in vitro methods.

Unfortunately, the manuscript could be not accepted for publication in this form due to following mistakes:

1. For determination of degree of substitution (DS) authors of (1) are used the correlation of integration areas of carbon atoms of methyl groups and C2 atom of furanose ring (lines 158-166). To my opinion it’s a wrong idea due to big differences in relaxation time of carbon atoms with different chemical environments. Much more better results of DS may be obtained using 1H NMR spectroscopy.

(1) Thanks for your suggestions. The method of calculating degree of substitution (DS) was based on the integration areas of carbon atoms, which was according to Chen and Zhang. (J. Zhang, W. Tan, Y. Mi, F. Luan, L. Wei, Q. Li, F. Dong and Z. Guo, Starch - Stärke, 2019, 71. Y. Chen, W. Tan, Q. Li, F. Dong, G. Gu and Z. Guo, Int J Biol Macromol, 2018, 113, 1273-1278.) We used DMSO-d6 as the solvent in this study. In the 13C NMR spectrum, the carbon absorption peaks of methyl groups (COOCH3) and C2 atom of furanose ring were separated from the other peaks even in the different chemical environments. So we think it was reasonable to determine the DS based on the integration areas of carbon atoms.

2. It is unclear how was calculated the DS for other compounds. The DS calculated using C/N ratio (table 1) is strongly differ for those presented in table 1. For example, the DS for compounds 2NS reveals 0.95 vs 0.30 in table 1.

(2) Thanks for your suggestions. The formulas for calculating the degree of substitution (DS) were added in the revised paper. Besides, we further purified the samples and did elemental analysis. The results were listed in Table 1 and we have revised the value of DS.

3. As a positive control was used carbendazim – compounds of different class benzimidazole. It should be explained.

(3) Thanks for your suggestions. The purpose of the positive control is to demonstrate the sensitivity of the test. We used carbendazim as the positive control because it has significant antifungal activity. And it is common to use carbendazim as a positive control in other papers. Shentu, X., Zhan, X., Ma, Z., Yu, X., Zhang, C. Brazilian Journal of Microbiology, 2014, 45(1), 248–254. Wang, L., Li, C., Zhang, Y., Qiao, C., Ye, Y.. Journal of Agricultural and Food Chemistry, 2013, 61(36), 8632–8640. So it is reasonable to use carbendazim as the positive control in the antifungal assay.

4. The inhibitory activities of 2NS, 3NS, 4NS are compared with inulin. However, to my opinions the inhibitory activities of compound 3 should be included because it is well known that amines and ammonium salts are effective antifungal agents.

(4) Thanks for your suggestions. We have added the inhibitory indices of 6-amino-6-deoxy-3,4-acetyl inulin (3) in Figure 5-7 and discussed the results in the section of 3.2.

5. Lines 165,166. “Protons” should be replaced for “carbons”.

(5) Thanks for your suggestions. We have corrected the word into carbons in the revised paper.

6. Line 173. The peak at 1.9 ppm could be not assigned to residual amino group. It should be proved. The 1H NMR spectra of compound 3 should be provided.

(6) Thanks for your suggestions. According to Ren, the peaks at 1.8-1.9 ppm were assigned to the amino groups using DMSO-d6 as the solvent. (J. Ren, P. Wang, F. Dong, Y. Feng, D. Peng and Z. Guo, Carbohydrate Polymers, 2012, 87, 1744-1748.) The amino groups were not completely substituted based on the 1H NMR spectra. Besides, the 1H NMR spectrum of compound 3 has been added in Figure 3.

Reviewer 2 Report

The manuscript Wei et al., entitled “Synthesis, characterization, and antifungal activity of Schiff bases of inulin bearing pyridine ring” has presented interesting research findings and indeed an area of current research importance. However, I have found some shortcomings in the manuscript and would like to suggest the authors take into consideration the following points before this article could be reconsidered for publication.

Major comments:

1-      This attempt to develop Schiff bases from inulin is not first, same authors have published the development and assessment (antifungal) of Schiff bases of inulin, Guo et al., 2014. They do have cited their article but I haven’t seen a commentary or review on that in their introduction. Please, it is strongly advised that the author should review relevant literature and distinctively write the novelty of the present work. 

2-      New Schiff bases of inulin were successfully developed and their antifungal ability was evaluated. However, the findings (Figures) have shown some visible difference among them and with the control but no attempt has been made to assess these differences using any statistical approach, therefore it is strongly advised to use any statistical approach (two-way ANOVA with posthoc test) to evaluate these differences whether the difference is significant or non-significant.

3-      The development of Schiff bases is primarily been confirmed Using FTIR and NMR. However, the results are only descriptive as no attempt has been made to discussed them in detail with reference to literature e.g. (Cérantola et al 2004), hence it is advised to authors to review their result and discussion section with a focus on discussion with reference to literature.  

Minor comments:

1-      Line 12, Please change “has good” to “has a good”.

2-      Line 17, 13C NMR or 1H NMR why or has been used. I think as both have been used for evaluation so should be and. Can I suggest changing it to 13C NMR and 1H NMR throughout the manuscript, please?

3-      Line 67-69, suggested rewriting the sentence.

4-      Line 78- 79, FTIR spectrometers doesn’t make any sense to me rather write FTIR spectra.

5-      Line 79- KBr Disks? Please, it advised to write the full name of kBr and also write a sentence on the method, how these were developed.

6-      Line 93, the reaction was stirred? I think it should be reactants.

7-      Line 93, why argon protection has been used, a line on this, please.

8-      Line 169; remove respectively as you are not writing a comparative sentence. It is strongly advised to proofread the article with care. 

Author Response

Reviewer 2

The manuscript Wei et al., entitled “Synthesis, characterization, and antifungal activity of Schiff bases of inulin bearing pyridine ring” has presented interesting research findings and indeed an area of current research importance. However, I have found some shortcomings in the manuscript and would like to suggest the authors take into consideration the following points before this article could be reconsidered for publication.

Major comments:

1. This attempt to develop Schiff bases from inulin is not first, same authors have published the development and assessment (antifungal) of Schiff bases of inulin, Guo et al., 2014. They do have cited their article but I haven’t seen a commentary or review on that in their introduction. Please, it is strongly advised that the author should review relevant literature and distinctively write the novelty of the present work.

(1) Thanks for your suggestions. We have added the relevant literature about Schiff bases of inulin and the novelty of the present work in the Introduction section.

2. New Schiff bases of inulin were successfully developed and their antifungal ability was evaluated. However, the findings (Figures) have shown some visible difference among them and with the control but no attempt has been made to assess these differences using any statistical approach, therefore it is strongly advised to use any statistical approach (two-way ANOVA with posthoc test) to evaluate these differences whether the difference is significant or non-significant.

(2) Thanks for your suggestions. The samples and concentrations were taken into account by using two-way ANOVA test and Turkey test to evaluate these differences. The results showed these differences were significant and discussed in the section of 3.2.

3. The development of Schiff bases is primarily been confirmed Using FTIR and NMR. However, the results are only descriptive as no attempt has been made to discussed them in detail with reference to literature e.g. (Cérantola et al 2004), hence it is advised to authors to review their result and discussion section with a focus on discussion with reference to literature.

(3) Thanks for your suggestions. We have discussed and analyzed the chemical structure of all the samples based on the relative references in the section of 3.1.

Minor comments:

1. Line 12, Please change “has good” to “has a good”.

(1) Thanks for your suggestions. We have corrected the sentence in the revised paper.

2. Line 17, 13C NMR or 1H NMR why or has been used. I think as both have been used for evaluation so should be and. Can I suggest changing it to 13C NMR and 1H NMR throughout the manuscript, please?

(2) Thanks for your suggestions. We have change ‘or’ to ‘and’ in the revised paper.

3. Line 67-69, suggested rewriting the sentence.

(3) Thanks for your suggestions. We have rewritten the sentence in the revised paper.

4. Line 78-79, FTIR spectrometers doesn’t make any sense to me rather write FTIR spectra.

(4) Thanks for your suggestions. We have revised the word ‘spectrometers’ to ‘spectra’.

5. Line 79-KBr Disks? Please, it advised to write the full name of KBr and also write a sentence on the method, how these were developed.

(5) Thanks for your suggestions. We have added the method and the full name of KBr in the section of 2.2.

6. Line 93, the reaction was stirred? I think it should be reactants.

(6) Thanks for your suggestions. We have revised the word ‘reaction’ to ‘mixture’.

7. Line 93, why argon protection has been used, a line on this, please.

(7) Thanks for your suggestions. Triphenylphosphine was used as a debromination reagent in the bromination reaction. Besides, triphenylphosphine is reductive. The argon would protect triphenylphosphine from being oxidized by oxygen in the air under the condition of heating.

8. Line 169; remove respectively as you are not writing a comparative sentence. It is strongly advised to proofread the article with care.

(8) Thanks for your suggestions. We have deleted the word ‘respectively’ and revised the paper carefully.

Round 2

Reviewer 1 Report

Authors have been performed huge job. The revised manuscript may be recommended for publications.